# Characteristics of Children and Adolescents with Hyperinsulinemia Undergoing Oral Glucose Tolerance Test: A Single-Center Retrospective Observational Study

**DOI:** 10.3390/diseases11030110

**Published:** 2023-08-28

**Authors:** Clelia Cipolla, Ilaria Lazzareschi, Antonietta Curatola, Claudia Lasorella, Lucia Celeste Pane, Linda Sessa, Giulia Rotunno, Donato Rigante, Giorgio Sodero

**Affiliations:** 1Department of Life Sciences and Public Health, Fondazione Policlinico Universitario A. Gemelli IRCCS, Largo Francesco Vito n. 1, 00168 Rome, Italy; clelia.cipolla@policlinicogemelli.it (C.C.); ilaria.lazzareschi@policlinicogemelli.it (I.L.); antonietta.curatola@guest.policlinicogemelli.it (A.C.); claudiaangela.lasorella@gmail.com (C.L.); luciaceleste.pane01@icatt.it (L.C.P.); linda.sessa01@icatt.it (L.S.); giulia.rotunno02@icatt.it (G.R.); donato.rigante@unicatt.it (D.R.); 2Università Cattolica Sacro Cuore, 00168 Rome, Italy

**Keywords:** hyperinsulinism, glycemic dysregulation, insulin resistance, diabetes mellitus, metabolic syndrome, obesity, oral glucose tolerance test, pediatric endocrinology, personalized medicine

## Abstract

The aim of this study was to evaluate a potential correlation between results of the oral glucose tolerance test (OGTT) and the auxological/metabolic parameters in a cohort of overweight patients assessed for suspicion of hyperinsulinism. We analyzed 206 patients, comparing those with insulin peak below (nonhyperinsulinemic) and over 100 uIU/mL (hyperinsulinemic) at the OGTT. We found a significant difference in weight (*p =* 0.037), body mass index (BMI, *p* < 0.001) and BMI standard deviations (SD, *p* < 0.001), waist circumference (*p* = 0.001), hip circumference (*p* = 0.001), and waist-to-height ratio (WHtR, *p =* 0.016) between the two groups. Analyzing the median insulin value during OGTT in the whole population, a weakly positive correlation emerged with weight SD (*p* < 0.001; *rho* = 0.292) and a moderate positive correlation with BMI SD (*p* < 0.001; *rho* = 0.323). We also found a weakly positive correlation with waist circumference (*p* = 0.001; *rho* = 0.214), hip circumference (*p =* 0.001; *rho* = 0.217), and WHTR (*p =* 0.016; *rho* = 0.209) and a moderate positive correlation with the HOMA index (*p* < 0.001; *rho* = 0.683). The median insulin value correlates with high triglyceride (*p* < 0.001; *rho =* 0.266) and triiodothyronine values (*p* = 0.003; *rho* = 0.193) and with low HDL values (*p* < 0.001; *rho =* −0.272). In clinical practice the interpretation of laboratory and anthropometric parameters could predict the level of insulin, highlighting also a possible underlying diagnosis of insulin resistance and/or hyperinsulinemia without performing an OGTT.

## 1. Introduction

Obesity is a complex chronic condition defined by an excess of body weight compared to height, with an increase in body mass index (BMI), subsequent dysfunction of adipose tissue, and unavoidable metabolic consequences [1]. It has been widely clarified from several studies that this disease affects not only the adult population, and indeed, the worldwide prevalence of obesity has increased to 47.1% in pediatric patients over the past 3 decades [2]. This situation has been further worsened in recent years with the severe acute respiratory syndrome coronavirus 2 (SARS-CoV-2) pandemic, which has led to significant changes in the lifestyle of children, reduction in physical activity and frequent adhesion to obesogenic diets [3]. It is also established that overweight and obesity lead to higher risk of metabolic consequences, which consist of metabolic syndrome, hyperinsulinism, and diabetes mellitus (DM), followed by a subsequently increased cardiovascular risk of these patients and reduced life expectancy [1].

Fasting plasma glucose (FPG) and 2 h plasma glucose following an oral glucose tolerance test (OGTT) are currently the gold standard for diagnosing a glycemic dysregulation in both children and adults [4]; in fact, OGTT consents to classify patients as having normal glucose tolerance, impaired fasting glucose, or impaired glucose tolerance, being prepathological conditions, or DM. In parallel with blood glucose assessment, plasma insulin determination also provides a chance to evaluate insulin peak and the presence of hyperinsulinism, which is largely associated with insulin resistance [5].

The aim of the present study was to evaluate any potential correlation between results of OGTT and the auxological or metabolic parameters of a cohort of overweight children assessed for suspected glucose dysregulation, highlighting differences in these parameters between those who had a final diagnosis of isolated hyperinsulinism or DM and those in whom these diagnoses were ruled out.

## 2. Patients and Methods

### 2.1. Study Characteristics

This is a single-center retrospective observational study evaluating the possible correlation between glycemia and insulin peaks during OGTT and both auxological and metabolic parameters of children displaying a suspicion of either DM or hyperinsulinemia. A formal ethics committee approval was not requested since the General Authorization to Process Personal Data for Scientific Research Purposes (authorization no. 9/2014) declared that retrospective archive studies that use ID codes, preventing the data from being traced back directly to data subjects, do not need a formal ethics approval [6]. Patients’ parents were informed about the aims of the study and signed a written informed consent for processing their anonymized personal data, allowing full access to the children’s medical records.

### 2.2. Patients’ Selection

We collected and analyzed auxological and metabolic data from the medical records of all overweight children and adolescents who underwent OGTT at our hospital (*Fondazione Policlinico Universitario A. Gemelli IRCCS*) for the clinical suspicion of glycemic dysregulation, i.e., hyperinsulinemia and DM, over the period from February 2015 to March 2023.

The initial screening allowed a preselection of 215 children evaluated at the pediatric endocrinology outpatients’ clinic; after this first selection, we retrieved clinical and laboratory data of patients who subsequently underwent OGTT for a total of 206 patients (81 males and 125 females, with an age range from 6 to 18 years).

All the remaining patients, despite a clinical suspicion, were lost at the follow-up (they refused to be tested or decided to be followed up at other hospitals). The medical records were retrospectively analyzed and the information on auxological parameters (weight, height, BMI, and waist and hip circumference) and metabolic parameters (lab tests performed before the stimulus test, i.e., thyroid-stimulating hormone [TSH], thyroxine [fT_4_], triiodothyronine [fT_3_], glycemia, insulin, transaminases, bilirubin, insulin growth factor-1 [IGF-1], glycated hemoglobin [Hb1Ac], and lipid profile) and those relating to the OGTT (insulin and glycemia peaks) were collected. We also analyzed the possible correlation with three indirect indices derived from the parameters previously reported: waist–hip ratio (WHR), waist-to-height ratio (WHtR), and homeostatic model assessment for insulin resistance (HOMA-IR). The first two parameters can be used to indirectly quantify the cardiovascular risk of patients [7], while the HOMA-IR index is an indirect marker of insulin resistance (HOMA-IR = fasting glucose levels [mmol/L] × fasting insulin levels (μU/mL)/22.5 or fasting glucose levels [mg/dl] × fasting insulin levels (μU/mL)/405) [8].

All extracted data were collected in an Excel database by two of the authors; subsequently, the collected parameters were checked by the leading researcher. Finally, the database obtained was used for performing the statistical analysis.

### 2.3. Characteristics of the OGTT

The OGTT is the gold standard method for assessing glucose tolerance, as well as insulin sensitivity and secretion, in both children and adults [9]. A peripheral venous access is required for blood sampling and for any eventual emergency; the test should be performed in the morning after at least 3 days of unrestricted diet and without restriction on a patient’s physical activity; an overnight fast of 10–12 h, during which only the administration of water is allowed, should precede the test. After collection of the fasting blood sample, the patient is asked to drink glucose (1.75 g/kg of body weight, based on the ideal body weight, with a maximum dose of 75 g, as usually undertaken for adults) within five minutes [5]; the patient should also remain seated or lying down throughout the following 2 h of the test. A complete glucose tolerance test analyzes samples taken at 0, 30, 60, 90, and 120 min. Serial assessments of glucose and insulin allow the identification of patients with glycemic dysregulation and even, in some cases, the diagnosis of DM [9].

There is an important variability in the interpretation of insulin peaks during OGTT, with variable cut-offs based on the type of guidelines considered or on the experience of single centers [5]; currently, a defined cut-off value in the pediatric population has not yet been defined, and there are several possible cut-offs [10]:-The calculation of the sum of the insulin measurements at the different sampling times during the OGTT > or <2083.5 pmol/L (300 μU/mL);-An insulin peak ≥ 1041.75 pmol/L (150 μU/mL);-A blood insulin value ≥ 520.88 pmol/L (75 μU/mL) when sampled 120 min after glucose loading.

The cut-off value considered in our study (100 uIU/mL) corresponds to that commonly used in our department to identify hyperinsulinemic subjects.

### 2.4. Statistical Analysis

Categorical variables were reported as numbers and percentages. The normality of the distribution of continuous variables was verified using the Shapiro–Wilk test. Non-normally distributed continuous variables were reported as the median and interquartile ranges (IQR). Statistical comparisons between groups were obtained with the Chi-square test or Fisher’s exact test, as appropriate, for categorical variables and the Mann–Whitney U test for non-normally distributed continuous variables. Spearman’s correlation was used to establish the existence of correlations between the median insulin value during OGTT and other anthropometric and metabolic parameters.

We considered a two-tailed *p*-value less than 0.05 to be statistically significant. Statistical analysis was performed with IBM SPSS Statistics for Windows software (version 25.0, SPSS Inc., 7 Chicago, IL, USA).

## 3. Results

### 3.1. Patients’ Characteristics

We recruited a total of 206 children and adolescents admitted to the pediatric endocrinology outpatients’ clinic of the Fondazione Policlinico Universitario A. Gemelli IRCCS (in Rome). Out of these, 81 (39.3%) were males, and their overall median age was 12 years (IQR 9.75–15.00). Sixty-eight children were prepubertal (33%), and 138 had reached a full pubertal development (67%).

Thirteen patients (6.3%) were affected by definite pathologies (Borjeson–Forssman–Lehman syndrome, pituitary solid swelling, acute lymphoblastic leukemia, infantile desmoplastic ganglioglioma, previous medulloblastoma, pituitary microadenoma, Costello syndrome, retinitis pigmentosa, left cortical hemisphere dysplasia, diencephalic astrocytoma, germinoma, Kaufman syndrome, and high-grade glioma).

Forty-one patients (19.9%) had a BMI under the 95th percentile, while 165 (80.1%) had an increased BMI (>95th); comparing patients with BMI <95th (overweight) with those having BMI >95th (obese), no statistically significant differences emerged in the relationship with a family history of DM, positivity of anti-TG and anti-TPO antibodies, and insulinemic peak. A statistically significant difference was observed for gender (*p* < 0.01) and pubertal stage (*p* < 0.01).

Additionally, in the obese group (BMI > 95th), we found higher values of waist and hip circumference, WHtR, HOMAi, transaminases, TSH, fT_3_, IGF-1, and Hb1Ac and lower values of HDL. However, the comparison was performed on two samples with different numbers of subjects. All details are given in Table 1.

Out of all, 183 patients (88.83%) had a waist circumference higher than the 75th percentile, while only 22 patients (10.67%) were under this cut-off. All the anthropometric and demographic data of our population are shown in Table 2.

One hundred and twenty-five patients (60.7%) had a family history of DM. The mean blood glucose value detected in our study population was 71.26 ± 10.13 mg/dL. Both glucose and insulin median values during OGTT and all the other metabolic parameters analyzed have been summarized in Table 3. None of our patients had a subsequent diagnosis of DM.

Our study population was divided into two groups on the basis of the insulin peak during the OGTT, regardless of the sampling time:-Group A: patients with OGTT insulin peak below 100 uIU/mL, for a total of 73 patients, included in our nonhyperinsulinemic group;-Group B: patients with OGTT insulin peak over 100 uIU/mL, for a total of 133 patients included in our hyperinsulinemic group.

### 3.2. Main Findings

We analyzed any possible differences for patients with and without hyperinsulinemia according to our classification criteria. No statistical differences were highlighted for age, sex, height, degree of obesity, pubertal stage, and familial history of DM. A statistically significant difference was highlighted for weight (*p* = 0.037), weight standard deviations (SD, *p* = 0.001), BMI (*p* < 0.001), BMI SD (*p* < 0.001), waist circumference (*p* = 0.001), hip circumferences (*p* = 0.001), and WHtR (*p* = 0.016) (see Table 4).

Basal insulin levels were significantly different between the two groups, as expected from our classification criteria (group A 9.80 uIU/mL (IQR 7.2–13.95 uIU/mL); group B 19.60 uIU/mL (IQR 14.5–29 uIU/mL); *p* < 0.001). We reported also the same statistical correlation for the HOMA index, which was indirectly correlated with basal insulin results (group A 1.74 (IQR 1.08–2.45); group B 3.53 (2.44–5.07); *p* < 0.001).

Finally, from the analysis of the metabolic parameters performed in all patients, a statistically significant difference emerged for triglycerides and fT_3_, which were higher in group B subjects (*p* = 0.004 and *p* = 0.042, respectively). In contrast, patients in group A had a median HDL value of 49.5 (IQR 41.25–56) compared to 45 (IQR 37–52) for patients in group B (*p* = 0.03). All details are reported in Table 5 and summarized in Figure 1.

Analyzing the median insulin value during OGTT in the whole population, a weakly positive correlation emerged with weight SD (*p* < 0.001; *rho* = 0.292) and a moderate positive correlation with BMI SD (*p* < 0.001; *rho* = 0.323).

We also found a weakly positive correlation with waist circumference (*p* = 0.001; *rho* = 0.214), hip circumference (*p* = 0.001; *rho* = 0.217), and WHTR (*p* = 0.016; *rho* = 0.209) and a moderate positive correlation with the HOMA index (*p* < 0.001; *rho* = 0.683). As regards metabolic parameters, the median insulin value was associated with higher levels of both triglycerides (*p* < 0.001; *rho* = 0.266) and fT_3_ (*p* = 0.003; *rho* = 0.193) and with lower HDL values (*p* < 0.001; rho= −0.272).

## 4. Discussion

It is wellestablished that incidence of obesity has dramatically increased worldwide in the last decade, both in the adult and pediatric populations [2,11]: this phenomenon underwent a further acceleration as a post-acute sequela of the SARS-CoV-2 pandemic in parallel with the increased rate of other endocrinological diseases [3] following the bad eating habits and poor lifestyles recorded in that period [12]. In addition, the increased incidence of obesity has also led to higher prevalence of DM, hyperinsulinemia, and insulin resistance [1], so that the identification of early markers of hyperinsulinemia might be crucial and could lead, starting from the pediatric age, to effective primary prevention campaigns aiming to reduce onset and progression of overweight and obesity but also to secondary prevention strategies, treating subjects with hyperinsulinemia and reducing their risk of cardiovascular diseases and other complications in the long term [13].

OGTT is the most widely used method for assessing glucose tolerance, insulin secretion, and the potential presence of insulin resistance; it has long been widely used both in clinical practice and for research purposes, although it may not be easy to perform, especially in younger and less compliant children [5].

In our study, we highlighted a correlation between glycemic and insulinemic peaks during OGTT and the auxological and metabolic parameters of overweight or obese children with a suspected diagnosis of DM or hyperinsulinemia. Our data, if confirmed by a future prospective study involving a larger cohort of patients, may help to more accurately select subjects who should undergo OGTT.

A relationship between insulin peak, weight, and BMI was expected. In fact, it is known that obese pediatric patients, even if not fulfilling the diagnostic criteria for DM and prediabetes, have already developed a condition of insulin resistance, partially reversible through proper dietary education and physical activity [14]. However, hyperinsulinemia could also affect patients with a normal weight, especially when a familial history of metabolic alterations is present, because of a genetic predisposition to metabolic syndrome and/or glucose dysregulation [15].

Sumano et al. performed a study on a small group of obese children with BMI >95th percentile, evaluating metabolic parameters in comparison with normal-weight subjects, and found that excessive BMI or increased waist circumference (>90th percentile) had a significant risk to produce a HOMA index pathologically increased, confirming its association with insulin levels [16].

The relationship between hyperinsulinemia and overweight is therefore bidirectional: Joslowski et al. analyzed a cohort of 91 obese young adults finding that, after a follow-up of 120 days, patients with higher insulin demand caused by a less balanced diet had a minor weight reduction; the authors hypothesized that a lower insulin demand related to food intake could facilitate weight loss, but their statistical analysis did not confirm a correlation between insulin and body fat mass [17].

Any pathological increases in insulin peak could be also associated with waist/hip circumferences and other indirect indices of overweight, like WHtR. For adults, waist circumference is commonly considered pathological when >88 cm in women and >102 cm in men [18]; in children, a value higher than the 90th percentile is commonly used as a cut-off to define overweight patients [18,19]. Moreover, WHtR is considered pathological when >0.5 cm, without differences between adults and children [20]. This cut-off correlates with an abnormal accumulation of adipose tissue in the abdomen, causing also ectopic accumulation of triglycerides, alterations in the lipid profile, and increased cardiovascular risk [21]. For this reason, these parameters are now accepted as markers of metabolic dysregulation [22] and may suggest an underlying hyperinsulinemia.

In addition to WHtR, there are currently several indices that could estimate the presence of insulin resistance or DM, and many of them are based on glucose and triglycerides, like the triglyceride–glucose index (TyG = Ln (fasting TG (mg/dL) × FPG (mg/dL)/2)) [23]. However, these ones are only validated for adults, and their available evidence for the pediatric population is partial. We did not calculate these indeces in our patients, but regardless of that, we were unable to confirm a statistical correlation with total and LDL cholesterol.

We found a significant association between insulin peak and triglycerides: this association was expected because hypertriglyceridemia is a key criterion for the diagnosis of metabolic syndrome [24], but we were surprised not to confirm a significant *p*-value for the remaining lipid profile. We could hypothesize that triglycerides are more influenced by diet and caloric intake than cholesterol; in fact, total cholesterol could also be increased in particular genetic conditions, like familial hypercholesterolemia, independently from overweight. Our findings were not in line with other studies, like the analysis conducted by Khan et al. [25]: they observed statistically significant differences in LDL and total non-HDL cholesterol between adults with and without metabolic syndrome without performing a subgroup analysis keeping patients’ BMI in consideration. However, this study was solely conducted on adult patients, and it is known that lipid levels can change during aging and during the pubertal progression of children [26].

We also highlighted an inverse correlation between hyperinsulinemia and HDL, as expected. These lipoproteins transport cholesterol from the peripheral cells to the liver, and their blood levels could increase in healthy subjects with moderate physical activity [27]; a reduction in periodic exercise could consequently lead to a reduction in HDL, increased body weight and BMI, and also hyperinsulinism [1].

Finally, we found a correlation between hyperinsulinemia and fT_3_; Lei et al. highlighted this association in a retrospective study involving 91 children with a previous diagnosis of Hashimoto’s thyroiditis, finding higher levels of HOMA in patients with confirmed hypothyroidism compared to those having a normal thyroid function [28]. Despite the statistical association, our population had, in general, higher levels of fT_3_, and this finding is partially explained by the increased deiodinase activation during childhood caused by the growth process; this enzyme could also be activated during different inflammatory processes and might be related to lipid accumulation [29].

A limitation of our study is the small sample size of children and adolescents analyzed; it is possible that an analysis conducted on a larger cohort might have led to different results, confirming alterations that in our analysis were not statistically significant. For example, we did not find a difference in serum urate between hyperinsulinemic and nonhyperinsulinemic subjects, while Niu et al. described the possible mediating role of uric acid in insulin resistance in a cohort of 369 obese children [30]. In another cross-sectional study involving 245 children, it was found that a small increase (1 mg/dL) in urate could increase the chance of insulin resistance (91%) [31]. Our classification into hyperinsulinemic and nonhyperinsulinemic groups is based on an arbitrary insulin peak cut-off, and therefore, a different classification method could probably lead to different results.

Finally, the comparison performed between patients with BMI lower and higher than 95th is not homogeneous because of the different size of the two samples. Our statistical analysis did not show significant differences in the insulin peak between the two groups, and this is partially explained by the different size of samples. A sub-analysis conducted on the absolute value of the BMI showed a statistically significant correlation with the insulin peak: as we expected, BMI is in fact higher in subjects with insulin peak >100; therefore, it is possible that a higher number of patients with BMI <95th could get different results.

In conclusion, the management of the obese pediatric patient should be global, and it is essential to analyze all possible direct and indirect markers of hyperinsulinemia to identify subjects who need to undergo an OGTT.

## 5. Conclusions

In our cohort, we highlighted a statistically significant correlation between patients with hyperinsulinemia and other metabolic/anthropometric parameters such as weight, BMI, waist circumference, hip circumference, WHtR, HDL, triglycerides, IGF-1, and fT_3_. Our results, if confirmed by further prospective studies with a larger sample size, could suggest the use of these parameters to guide physicians toward the diagnosis of glycemic dysregulation and hyperinsulinism in order to reduce the use of the OGTT test, which will be performed as a confirmation tool only in the most proper cases. In addition, for clinical practice, the interpretation of laboratory and anthropometric parameters could estimate with good approximation the level of insulin, highlighting also a possible underlying diagnosis of insulin resistance and/or hyperinsulinemia.

## Figures and Tables

**Figure 1 diseases-11-00110-f001:**
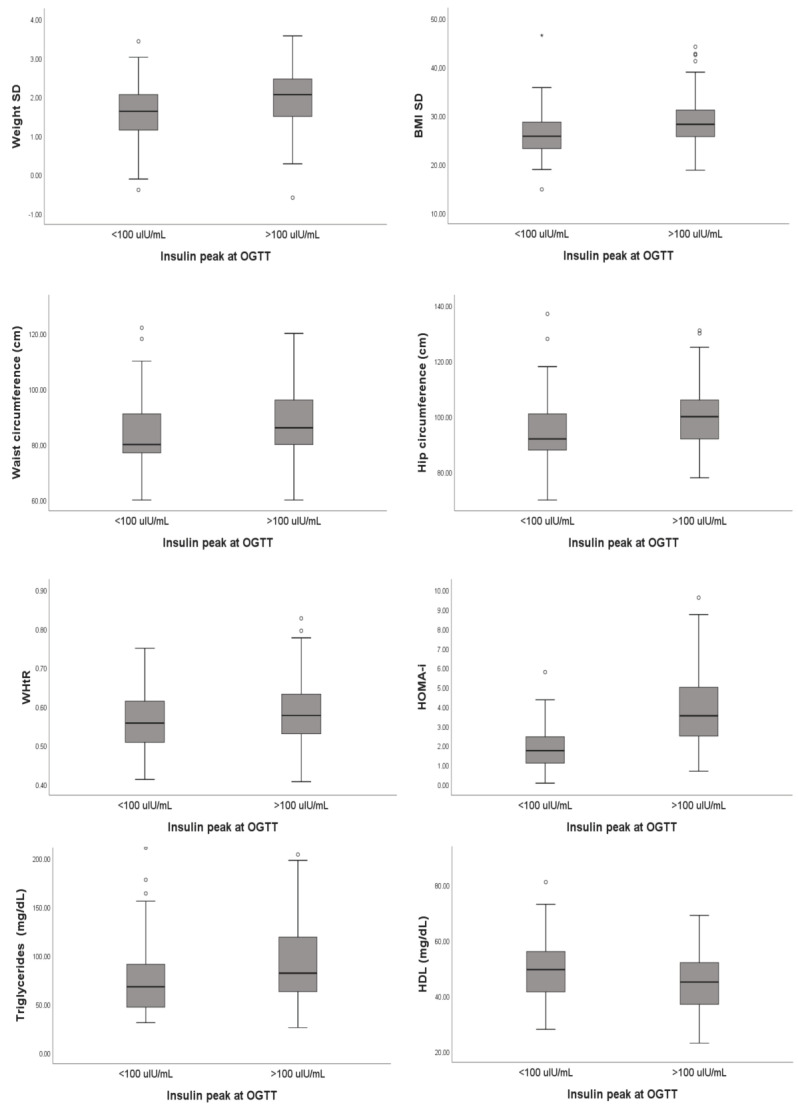
Summary of the correlations highlighted during our statistical analysis.

**Table 1 diseases-11-00110-t001:** Comparison between patients with BMI lower than 95th and higher than 95th percentile.

	BMI < 95th	BMI > 95th	*p* Value
Waist circumference (cm)	79 [74.5–85.5]	88 [80–97]	>0.001
Hip circumference (cm)	93 [88.5–97.5]	100 [90–106]	0.001
Waist/hip circumference	0.86 [0.82–0.89]	0.90 [0.87–0,94]	>0.001
WHR (cm)	0.51 [0.47–0.53]	0.58 [0.54–0.63]	>0.001
HOMAi	2.32 [1.48–3.64]	2.37 [2.01–4.43]	0.009
TSH (μIU/mL)	1.96 [1.34–2.62]	2.63 [1.97–3.57]	>0.001
fT_3_ (pg/mL)	3.6 [3.4–4]	4.1 [3.3–4.5]	>0.001
IGF-1 (ng/mL)	301 [233.5–374]	215 [163–315]	0.002
Hb1Ac (mmol/mol)	35 [32–34]	36 [34–32]	0.006
HDL (mg/dL)	49 [43.25–58]	46 [38–53]	0.006

**Table 2 diseases-11-00110-t002:** Anthropometric and demographic characteristics of patients recruited in our study.

	Median (IQR)	N (%)
Age	12.00 [9.75–15.00]	
Sex (M)		81 (39.3)
Pubertal development -Prepubertal -Full pubertal development		68 (33) 138 (67)
Familial history of diabetes		125 (60.7)
Weight (kg)	64.00 [51.22–76.85]	
Weight SD	1.94 [1.35–2.35]	
Height (cm)	153.00 [142.82–162.22]	
Height SD	0.32 [−0.56–1.01]	
BMI (kg/m^2^)	27.53 [24.82–30.19]	
BMI classification - Overweight (<95th) - Obese (>95th)		41 (19.9) 165 (80.1)
BMI SD	2.01 [1.67–2.31]	
Waist circumference (cm)	85.00 [79.00–95.00]	
Waist circumference SD >75th		183 (88.93%)
Hip circumference (cm)	98.00 [89.75–105.0]	
WHR (cm)	0.89 [0.85–0.93]	
WHtR (cm)	0.56 [0.52–0.62]	

**Table 3 diseases-11-00110-t003:** Metabolic parameters of patients recruited in our study.

	Median (IQR)	N (%)
Insulin (μIU/mL)	16.05 [10.80–22.77]	
HOMA-I	2.67 [1.89–4.26]	
GOT (IU/L)	22.00 [19.0–25.0]	
GPT (IU/L)	19.00 [15.0–25.0]	
Bilirubin (mg/dL)	1.00 [1.0–2.0]	
TSH (μIU/mL)	2.42 [1.80–3.49]	
fT_3_ (pg/mL)	4.10 [3.70–4.40]	
fT_4_ (pg/mL)	11.30 [10.50–12.40]	
Anti-TG antibodies (IU/mL)		15 (7.3)
Anti-TPO antibodies (IU/mL)		7 (3.4)
IGF-1 (ng/mL)	234.00 [169.5–336.5]	
IGF-1 SD	0.17 [−0.55–1.11]	
Urate (mg/dL)	4.00 [4.0–5.50]	
C-reactive protein (mg/L)	4.00 [1.0–8.50]	
Hb1Ac (mmol/mol)	36.00 [34.0–38.0]	
Total cholesterol (mg/dL)	153.00 [138.0–171.0]	
HDL (mg/dL)	46.00 [39.0–54.0]	
LDL (mg/dL)	89.00 [76.0–102.0]	
Triglycerides (mg/dL)	78.50 [55.0–110.5]	
OGTT-basal glucose (mg/dL)	84.00 [80.0–89.0]	
30 min	116.00 [104.0–134.0]	
60 min	114.50 [100.0–131.75]	
90 min	107.00 [95.0–122.0]	
120 min	105.00 [95.0–116.50]	
OGTT-basal insulin (μIU/mL)	16.05 [10.70–22.77]	
30 min	95.20 [50.60–159.55]	
60 min	89.70 [56.15–158.0]	
90 min	71.30 [46.15–131.85]	
120 min	79.30 [47.35–123.85]	
Insulin peak -<100 μIU/mL ->100 μIU/mL		73 (35.4) 133 (64.6)

**Table 4 diseases-11-00110-t004:** Statistical analysis of anthropometric data related to patients recruited in our study.

	Group A	Group B	*p* Value
Age (median [IQR])	12.00 [9–16.5]	12.00 [10–14]	0.502
Sex (M), n (%)	29 (39.7)	52 (39.1)	0.930
Weight classification n (%) -Overweight - Obese	18 (24.7) 55 (75.3)	23 (17.3) 110 (82.7)	0.205
Pubertal classification n (%) -Prepubertal - Full pubertal development	25 (34.2) 48 (65.8)	43 (32.3) 90 (67.7)	0.780
Familial history of diabetes, n (%)	45 (61.6)	80 (60.2)	0.834
Weight (kg) (median [IQR])	61.00 [48.75–70.90]	66.00 [54.05–80.5]	0.037
Weight SD (median [IQR])	1.63 [1.14–2.07]	2.06 [1.47–2.46]	0.001
Height (cm) (median [IQR])	155.40 [140.15–163]	152.60 [144.25–161]	0.951
Height SD (median [IQR])	0.15 [−0.75–0.79]	0.42 [−0.43–1.13]	0.115
BMI (kg/m^2^) (median [IQR])	25.78 [23.11–28.68]	28.23 [25.62–31.24]	<0.001
BMI SD (median [IQR])	1.86 [1.56–2.06]	2.08 [1.79–2.36]	<0.001
Waist circumference (cm) (median [IQR])	80.00 [76.25–91]	86.00 [80–96.5]	0.001
Waist circumference SD >75th, n (%)	59 (80.8)	124 (93.9)	0.004
Hip circumference (cm) (median [IQR])	92.00 [87–101.5]	100.00 [92–106]	0.001
WHR (cm) (median [IQR])	0.89 [0.84–0.93]	0.89 [0.86–0.93]	0.617
WHtR (cm) (median [IQR])	0.55 [0.50–0.61]	0.57 [0.52–0.63]	0.016

**Table 5 diseases-11-00110-t005:** Statistical analysis of the metabolic parameters related to patients recruited in our study.

	Group A	Group B	*p* Value
Glucose (mg/dL) (median [IQR])	69.00 [63–78]	72.00 [66–78.5]	0.278
Insulin (uIU/mL) (median [IQR])	9.80 [7.2–13.95]	19.60 [14.5–29]	<0.001
HOMA-i (median [IQR])	1.74 [1.08–2.45]	3.53 [2.44–5.07]	<0.001
GOT (IU/L) (median [IQR])	22.00 [19–26]	21.00 [18–25]	0.525
GPT (IU/L) (median [IQR])	19.00 [14.25–24]	19.00 [15–27]	0.568
Total bilirubin (mg/dL) (median [IQR])	1 [1–1]	1 [1–2.5]	0.617
TSH (uIU/mL) (median [IQR])	2.30 [1.78–3.10]	2.67 [1.80–3.57]	0.089
fT_3_ (pg/mL) (median [IQR])	4.00 [3.6–4.2]	4.1 [3.7–4.5]	0.042
fT_4_ (pg/mL) (median [IQR])	11.60 [10.45–12.6]	11.25 [10.5–12.1]	0.199
Anti-TG antibodies (IU/mL) (n, %)	6 (8.2)	9 (6.8)	0.701
Anti-TPO antibodies (IU/mL) (n, %)	1 (1.4)	6 (4.5)	0.425
IGF-1 (ng/mL) (median [IQR])	197.00 [158–277.5]	251.5 [172.5–365.7]	0.010
IGF-1 SD (median [IQR])	0.09 [−1.17–0.91]	0.31 [−0.49–1.23]	0.127
Urate (mg/dL) (median [IQR])	4 [3–4]	4 [4–6]	0.264
C-reactive protein (mg/L) (median [IQR])	2	4.5 [1–8.25]	0.864
Hb1Ac (mmol/mol) (median [IQR])	35.5 [33–37]	36.00 [34–38]	0.332
Total cholesterol (mg/dL) (median [IQR])	156.00 [138–170]	152.5 [138–171.25]	0.678
HDL (mg/dL) (median [IQR])	49.5 [41.25–56]	45.00 [37–52]	0.003
LDL (mg/dL) (median [IQR])	90.00 [76–101.75]	89.00 [76.25–103]	0.887
Triglycerides (mg/dL) (median [IQR])	68.00 [47–91]	82.00 [62–102]	0.004

## Data Availability

No datasets were generated and analyzed during the study.

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
