# Peer review of "Characteristics of Children and Adolescents with Hyperinsulinemia Undergoing Oral Glucose Tolerance Test: A Single-Center Retrospective Observational Study"

_diseases, 2023, doi:10.3390/diseases11030110_

Round 1

Reviewer 1 Report

The article gives data on children and adolescents to find if it is possible to discover children with hyperinsulinemia without doing an OGTT.  They found that weight, BMI, waist circumference, hip circumference and waist to height ratio were significant different between two groups with either an insulin peak under or above 100uIU/ml. These findings might indicate that it is not necessary to do an OGTT when the children belong to group A. But how do you decide in an individual case to what group it belongs A or B? That is not made clear in your paper. Do you need more cases for that? The idea is excellent of course, but how do you do it in practice?

Author Response

Dear Editor,

first of all I would like to thank you for this opportunity of cooperation and for the opportunity to improve the overall quality of the manuscript entitled “Characteristics of children and adolescents with hyperinsulinemia undergoing oral glucose tolerance test: a single-centre retrospective observational study” for the potential publication upon “Diseases”.

This is a list of all amendments made in our manuscript in response to the two reviewers.

Reviewer 1

The article gives data on children and adolescents to find if it is possible to discover children with hyperinsulinemia without doing an OGTT.  They found that weight, BMI, waist circumference, hip circumference and waist to height ratio were significant different between two groups with either an insulin peak under or above 100 uIU/ml. These findings might indicate that it is not necessary to do an OGTT when the children belong to group A. But how do you decide in an individual case to what group it belongs A or B? That is not made clear in your paper. Do you need more cases for that? The idea is excellent of course, but how do you do it in practice?

Dear Reviewer,

thank you for your comment which helped us to improve our article.

We think that in clinical practice an alteration of the parameters analysed, with particular attention to body mass index (BMI), could identify the presence of a pathological insulin peak; our parameters are complementary to the OGTT and cannot replace it for the moment. However, they can suggest the diagnosis of hyperinsulinemia, selecting more accurately the subjects to test and allowing also an early initiation of therapy (physical activity, diet with reduced carbohydrate intake, and possible therapy with metformin).

Further prospective studies including a larger number of children and adolescents are needed to confirm these correlations, especially in patients with an increased BMI.

We also extended our statistical analysis by performing a further comparison of patients with BMI over and under the 95th percentile.

We hope you will find appropriate the changes we have made in our paper.

Reviewer 2 Report

Cipolla et al present a retrospective study in which clinical data of a medium-sized cohort of children and adolescents are scrutinized to determine whether laboratory and anthropometric parameters can be predictive of hyperinsulinaemia.

 Lines 195-201 of the manuscript underscore the importance of applying preventive measure towards the development of childhood obesity. It is shown here that several clinical and anthropometric measures are, in the tested population of hyperinsulinaemic children/adolescents, correlated to the insulin peaks and hyperglycemic curves observed during an OGTT.

It is concluded that the knowledge of some clinical and anthropometric values may substitute to (or complement) the performance of an OGTT in a young population.

I have a major issue related to this manuscript, that must be attended by the authors: in the study, approximately 20 % of the participants experience a milder obesity, in terms of BMI percentile. A comparison among these two groups (< 95th percentile and above the 95th percentile), although asymmetric in terms of number of participants, is essential and would be certainly interesting to the reader. Therefore, I call upon more analysis (from the existing data) from the authors. Indeed, on one side the manuscript presents correlation studies between clinical metabolic values and anthropometric measures with Insulin levels (supporting the main conclusion that an OGTT with glucose and insulin measures might be replaced by the above mentioned parameters). On the other side, as the BMI is a strong predictor of metabolic health, the investigation of whether the belonging of the study subjects above, or below, the 95th percentile of BMI has an impact on study results is mandatory.

Minor issue to be considered: some care should be taken in the presentation of figure 1: horizontal lines parallel to the X-axis may be removed and text of the graphs should be rendered with bigger fonts to ensure the reproduction in the final PDF is easily legible.

just provide a final proofreading.

Author Response

Dear Editor,

first of all I would like to thank you for this opportunity of cooperation and for the opportunity to improve the overall quality of the manuscript entitled “Characteristics of children and adolescents with hyperinsulinemia undergoing oral glucose tolerance test: a single-centre retrospective observational study” for the potential publication upon “Diseases”.

This is a list of all amendments made in our manuscript in response to the two reviewers.

Reviewer 2

Cipolla et al present a retrospective study in which clinical data of a medium-sized cohort of children and adolescents are scrutinized to determine whether laboratory and anthropometric parameters can be predictive of hyperinsulinaemia.

Lines 195-201 of the manuscript underscore the importance of applying preventive measure towards the development of childhood obesity. It is shown here that several clinical and anthropometric measures are, in the tested population of hyperinsulinaemic children/adolescents, correlated to the insulin peaks and hyperglycemic curves observed during an OGTT.

It is concluded that the knowledge of some clinical and anthropometric values may substitute to (or complement) the performance of an OGTT in a young population.

I have a major issue related to this manuscript, that must be attended by the authors: in the study, approximately 20% of the participants experience a milder obesity, in terms of BMI percentile. A comparison among these two groups (<95th percentile and above the 95th percentile), although asymmetric in terms of number of participants, is essential and would be certainly interesting to the reader. Therefore, I call upon more analysis (from the existing data) from the authors.

Indeed, on one side the manuscript presents correlation studies between clinical metabolic values and anthropometric measures with Insulin levels (supporting the main conclusion that an OGTT with glucose and insulin measures might be replaced by the above mentioned parameters).

On the other side, as the BMI is a strong predictor of metabolic health, the investigation of whether the belonging of the study subjects above, or below, the 95th percentile of BMI has an impact on study results is mandatory.

Minor issue to be considered: some care should be taken in the presentation of figure 1: horizontal lines parallel to the X-axis may be removed and text of the graphs should be rendered with bigger fonts to ensure the reproduction in the final PDF is easily legible.

Dear Reviewer,

we greatly appreciate your comments on our work, which allowed us to improve it.

The comparison of the two groups (BMI > and < 95th percentile) had not been reported in the text in consideration of the asymmetry of the two groups (80% vs 20% of our sample).

Indeed, the analysis of the two groups with different BMI is the most interesting of the anthropometric measures; we therefore decided to integrate our statistical analysis with this information, also adding a new table (Table 1) within the article.

Insulin peak was not statistically different between the two groups and this is partly explained by the asymmetry in sample size; our statistical analysis has in fact shown that patients with an insulin peak >100 have a higher BMI value. Of course, it is possible that an analysis with a higher number of patients with BMI under the 95th percentile might have led to different results.

We have also revised the part of the discussion integrating the BMI data, when reported in the articles cited.

Thank you also for your comment related to our figure: we have made the requested changes.

We hope you’ll find appropriate the changes we have made to our paper.

Therefore, after these amendments made, we sincerely hope that you can consider valid such changes.

We are waiting for the final decision related to our manuscript.

Sincerely,

Giorgio Sodero & coauthors

Round 2

Reviewer 1 Report

I have read the revised version and I agree with publication, It is much clear now.

Reviewer 2 Report

I have no futher comments for the authors.

non applicable.